# Needs of Alzheimer’s Charges’ Caregivers in Poland in the Covid-19 Pandemic—An Observational Study

**DOI:** 10.3390/ijerph18094493

**Published:** 2021-04-23

**Authors:** Jagoda Rusowicz, Krzysztof Pezdek, Joanna Szczepańska-Gieracha

**Affiliations:** 1Department of Physiotherapy, University School of Physical Education, 51-612 Wrocław, Poland; joanna.szczepanska@awf.wroc.pl; 2Department of Physical Education and Sport Sciences, University School of Physical Education, 51-612 Wroclaw, Poland; krzysztof.pezdek@awf.wroc.pl

**Keywords:** dementia, elderly, social support, stress, caregiver burden

## Abstract

In Poland, 92% of elderly people with dementia are cared for at home from diagnosis until death, and 44% of caregivers provide care on their own, without any support from other people. The aim of this study was to identify the needs, created because of the Covid-19 pandemic, of caregivers of people with Alzheimer’s disease (AD). The study group consisted of 85 caregivers in the age range from 23 to 78 years and 80 (91.1%) were women. The questionnaire on the life situation of the caregiver and 10-item Perceived Stress Scale (PSS-10) were used. High levels of stress were found in 75 of the 85 subjects, representing 88% of the total. The greatest difficulties were identified in health care and in finding additional care for the charge. PSS-10 correlated with the deterioration of illness during Covid-19, changes in daily functioning, and concerns about both the health of the charge and caregiver. The level of stress severity in the caregiver group of charges with mild AD was higher than in the caregiver group of charges with moderate AD. The provision of extra care and professional psychological support for caregivers were identified as the greatest needs.

## 1. Introduction

Dementia is a current and serious public health problem in developed countries. It is one of the main causes of disability and dependency among older people worldwide [1]. As defined by the World Health Organization (WHO), it “(…) is a syndrome, usually of a chronic or progressive nature, in which there is a deterioration in cognitive function (i.e., the ability to process information) beyond what can be expected from normal ageing. Memory, thinking, orientation in time and place, visual and spatial orientation, understanding, calculation, learning ability and language functions are affected. Cognitive impairment is often accompanied and sometimes even preceded by a deterioration in emotional control, social behavior or motivation” [2]. Worldwide, around 50 million people have dementia, with nearly 60% living in low- and middle-income countries. AD is present in 60–70% of all dementia charges [2,3,4].

Poland is one of the fastest ageing countries in Europe, which results in increasing problems for the care of elderly members of society. Estimates show that over 500,000 people in Poland suffer from dementia, including over 300,000 from AD. It is worrying that only 15–20% of these charges have been diagnosed and undergo treatment. Additionally, it is known that within the next 20–25 years the number of charges will double, similar to other countries [3,4,5]. AD is a serious challenge for the Polish health care and social welfare system. Neurologists, psychiatrists, and geriatrists are responsible for diagnosing and treating the disease. The report of the Supreme Audit Office shows that the availability of geriatric care in Poland is insufficient. There is not even one such specialist per 100 thousand inhabitants (ratio is 0.8). For comparison, in the Czech Republic, the ratio is 2.1, in Slovakia 3.1, and in Sweden almost 8. Dementia, especially AD, is one of the costliest diseases of modern Europe. The costs generated by this disease include direct costs (treatment and nursing care, daily care at home), as well as those that can be described as indirect (loss of professional and social productivity of the charge and caregiver). According to the report by Alzheimer’s Disease International, costs rose to $604 billion in 2010 and reached $817 billion in 2015 [4,6]. In 2016, in the United States, informal care amounted to 18.2 billion hours, translating to US$230.1 billion [7].

Caregivers experience much more than high financial expenses. There are few diseases that engage in the long-term care of their family members more than AD [8]. In Poland, 92% of elderly people with dementia are cared for at home from diagnosis until death, and 44% of caregivers provide care on their own, without any support from other people [9,10]. It follows that the care of a charge with dementia rests mainly with caregivers who come from or are engaged by the family. The most numerous group of caregivers in Poland are spouses, who are close to the charge’s age (≥65 years). In Poland, as in the rest of the world, a greater proportion of people caring for charges with dementia are women [8,10,11].

This responsibility and care for the charge creates multiple burdens in the mental, physical, economic and social areas [12,13]. The mental burden increases over many years of care and accumulates, which can lead to mental burnout syndrome, for the caregiver. It also occurs as a result of living under permanent stress and the caregiver’s failure to cope with their own numerous negative emotions. It is estimated that 70% of caregivers suffer from permanent stress and 50% from depression and depression syndromes [8]. Physical burdens are a result of the caregiver taking over additional duties related to daily care of the charge and managing the household. These activities are often performed in the absence of cooperation from the charge and require overcoming their resistance.

As the illness progresses and the charge gradually loses the ability to self-serve, the physical burden continues to increases [8,14,15]. The economic burden is caused by the high cost of treatment and the provision of a professional caregiver or nurse. The social burden is understood as various forms of social isolation of the charge and caregiver. Spouse caregivers and adult children caregivers experience the greatest burden compared with other informal caregivers of people with dementia [16]. The lack of understanding of the disease and the appearance of socially unacceptable and sudden behavior in the charge hinders the normal functioning of the whole family. The long-term care provided by the caregiver deprives them of many opportunities to establish social contacts and limits their maintenance [8,17]. In addition, the prevalence of abuse risk, caused by anxiety and feelings of burden, is high among family caregivers [18].

The Covid-19 pandemic forced everyone to adapt to new rules of social functioning. It drastically changed the reality in which we live. It is impossible not to notice that the elderly and those suffering from chronic diseases were in the at-risk group. In the current situation, the caregivers of people with AD living in Poland also face a new reality and new problems. Behind every sick person there is a caregiver who struggles to take care of the sick family member every day.

The aim of this study was to identify the needs, which were created as a result of the Covid-19 pandemic, of caregivers of people with AD. Recognition of the needs and understanding of the situation of caregivers involves the possibility of providing real support, both physically, mentally, and socially, to caregivers and their charges.

## 2. Materials and Methods

### 2.1. Design of the Study

The observational study was conducted in a group of caregivers of AD charges (85 participants) living in Poland who met the following inclusion criteria:Providing care for AD charges before and during the Covid-19 pandemic,Completion of an online questionnaire assessing the situation of caregivers and the needs of care during a pandemic, andConsent to participate in the study, which means assessing the stress level using the 10-item Perceived Stress Scale.

The study lasted 3 months and was conducted in accordance with the Helsinki Declaration.

### 2.2. Participants

The study group consisted of 85 caregivers in the age range from 23 to 78 years old with an average age of 51 (±11.9) and 80 (91.1%) were women. The research group consisted of caregivers of charges with AD living in Poland who, having familiarized themselves with the information published on the Internet, decided to participate in the study on a voluntary basis and met the inclusion criteria. The characteristics of the study group and charges are presented in Table 1.

## 3. Measuring Instruments

### 3.1. Questionnaire on the Life Situation of the Caregiver

The questionnaire on the life situation of the caregivers was completely anonymous and was developed for this study. It consisted of two sections. The first section contained 20 questions about the life situation of the caregiver and information related to the needs of daily care and functioning for a charge with AD before and during the Covid-19 pandemic. The second section was the 10-item Perceived Stress Scale (PSS-10). Participation in the study was entirely voluntary.

It took no more than 10 min to complete the whole questionnaire. It did not require the respondents to share their personal data, medical records, or sensitive data, or require further contact. It was based on observations and feelings related to the new situation in which the respondents found themselves. The questionnaire used in the survey, translated from Polish, can be found in the Appendix A.

#### 10-Item Perceived Stress Scale (PSS-10)

The PSS-10 scale is used to test adults, both healthy and ill. The PSS-10 is a self-report instrument consisting of 10 items purported to assess “how unpredictable, uncontrollable, and overloaded respondents find their lives” [19]. Each of the items on the PSS-10 are rated on a 5-point Likert scale, ranging from 0 (never) to 4 (very often). The PSS-10 consisted of 6 positively and 4 negatively worded items. Negatively worded items were re-coded during analysis. Total scores ranged from 0 to 40, with higher scores indicating higher levels of perceived stress [20,21,22]. The PSS-10 is one of the most frequently used instruments to measure perceived stress [20]. Internal compatibility was checked in the study of a group of 120 adults, obtaining an alpha Cronbach index of 0.86. The correlation of all questions with the overall score was satisfactory. The reliability established on the basis of a double study of a group of 30 students at an interval of 2 days was 0.90 and at an interval of 4 weeks, was 0.72 [19].

### 3.2. Procedure of Data Collection

Due to the new rules of social functioning and the lack of opportunity to meet with participants during the Covid-19 pandemic, the study was conducted online. The form was made available at the beginning of August 2020 and concluded at the end of October 2020. The group selection was inherently random. The form settings allowed for the collection of a full set of responses (each respondent answered all questions).

The questionnaire was completed via the Internet (using a Google Docs form) by sending information about the study via e-mail and social networking sites associated with caregivers of Alzheimer’s charges from all over Poland, including open and private groups for caregivers on Facebook, portals, and information pages dedicated to dementia and elderly care. The invitation to participate in the study was also sent out via e-mail to organizations associated with caregivers of AD charges with a request to disseminate information among them. The biggest response came immediately after the information was posted in closed Facebook groups for caregivers.

### 3.3. Data Analyses

The study group was characterized using the following descriptive statistics: mean, standard deviation, minimum and maximum values, and, in the case of qualitative variables, numbers and percentages. Statistical tests were performed at a significance level of *p* < 0.05. Analyses were conducted using numerical tables, one-way analysis of variance (ANOVA), and non-parametric tools to investigate relationships between characteristics. The normality of the distribution of the continuous characteristics was determined using the Shapiro–Wilk test. The null hypothesis of normality of distribution was rejected for most characteristics of the study group. Therefore, the nonparametric Spearman correlation coefficient (ρ) was used to evaluate the interdependence between the characteristics (age of the caregiver, age of the charge, length of care, place of residence, and assistance received, among others). A one-way ANOVA with Scheffé’s post hoc test was performed to test whether the degree of dementia (comparisons of the three groups) influenced the scores on the PSS-10. In the Levene’s test of homogeneity of variance, there was no reason to reject the null hypothesis. The calculations were carried out using the STATISTICA 13.3 software of StatSoft.

## 4. Results

Before the pandemic, 41.2% of the respondents lived in a different place from their charge, 34.1% lived with their charge, while 24.7% lived together with their charge and other family members. At the time of the study, the percentages changed as follows: 42.4, 30.6, and 27.1%, respectively. The mean duration of illness was 6 years (SD ± 3.66), and the duration of care provided by a caregiver was 5 years (SD ± 3.74). The severity of dementia in AD charges before the epidemic is shown in Figure 1.

Deterioration of the AD charge’s health during Covid-19 was declared by 53% of caregivers, while deterioration of their own health by 40%. About 34% felt that their health had not deteriorated. Most respondents (69.4%) noted that the Covid-19 pandemic had changed their daily functioning. Notably, 78.8% of caregivers confirmed the emergence of new care needs for someone with AD during the Covid-19 pandemic. Caregivers were also asked about the areas in which problems emerged in the caregiver’s functioning in relation to the Covid-19 pandemic (respondents could mark up to 3 most important answers) from the proposed categories. An enumeration of the problems that arose with the time of Covid-19 are presented in Table 2. More than 83% of the respondents said they had not received any offer of help in the organizational, psychological, or social areas in relation to caring for someone with AD during the pandemic. Receiving help in at least one of these areas was declared by 16 caregivers (18.8%). Concerns about their charge’s health during the Covid-19 pandemic increased in 78.8% of caregivers. Caregivers’ concerns about their own health increased in 63.5% of respondents.

For the type of help that caregivers expect in relation to the difficult epidemiological situation, respondents could choose up to 3 of the most important types of assistance from the given categories. The responses are presented in Table 3.

### PSS-10

The mean PSS-10 score was 25.5 (± 4.94), with a minimum of 13 and a maximum of 39 points. The range of scores on the sten scale was 4–10 sten. Of the scores, 88% were between 7 and 10 sten, meaning high and very high levels of perceived stress, respectively (7 sten—18.8%, 8 sten—24.7%, 9 sten—28.2%, 10 sten—16.5%).

The data was analyzed for a correlation. There was no significant relationship between the PSS-10 score and the age of the caregiver, age of the charge, length of care, duration of dementia, place of residence, or degree of relationship to the charge. However, PSS-10 correlated with the deterioration of charge’s illness during Covid-19, deterioration of caregiver’s health during Covid-19, change in daily functioning, and concerns about the health of the charge with AD. Levels of perceived stress were associated with a definite worsening of the charge’s illness as perceived by the caregiver and a high degree of change in daily functioning, as well as a definite increase in the caregiver and a high degree of change in daily functioning, as well as a definite increase in concern for the charge’s health and a worsening of the caregiver’s health during pandemic. These correlations were statistically significant (*p* > 0.05). Table 4 presents the results of the correlation analysis.

A one-way ANOVA with Scheffé’s post hoc test was conducted to test whether the degree of dementia influenced the PSS-10 scores. Caregivers of charges with moderate dementia scored lower on the PSS-10 compared with caregivers of charges with mild or severe dementia. The largest and statistically significant differences occurred between the moderate dementia and severe dementia groups (*p* = 0.019 and effect power = 0.72.) (Figure 2).

## 5. Discussion

The study gave us an insight into the situation and needs of caregivers of charges with AD in Poland because of the Covid-19 epidemic. This is a very important issue in Poland because family caregivers are not recorded in the register, so it is difficult to establish details of their number or socio-economic situation. It is estimated that, in Poland, about 92% of older people with dementia live at home from the onset of the disease until death [15]. Clear problems were perceived by the interviewed caregivers in health care, understood as difficulties in accessing medical and nursing care or in buying medicines, as well as in finding additional care for a dependent person with AD.

In our study, most caregivers were children of the charge, with a predominance of women. The longer life expectancy of women is one reason for the prevalence of women as caregivers, as shown in other studies [10,18,23,24,25]. The second largest group of caregivers was daughter-in-law/son-in-law. This is a certain novelty in relation previous years, where the largest group were spouses [10,23,26]. Mazurek et al. [23] report that spouses are the most common caregivers, with children being the second most common group. It is likely that conducting the research through the medium of the Internet meant that we reached a younger generation of caregivers, children and son/daughter-in-law.

There are fundamental differences between spouse and child caregivers due to different generations. For example, adult child caregivers are more likely to work outside the home and experience scheduling conflicts. Spouse caregivers are more likely to experience physical limitations that make caregiving difficult and report more feelings of depression [17,27,28]. Our study group does not reflect the population of caregivers in Poland, but it does provide a new perspective on the problems of caregivers. We had almost equal representations of caregivers from large, medium, and small cities, as well as rural areas. This is important because caregivers in smaller towns and rural areas cannot rely on community-based initiatives, NGO-funded activities, and the work of various foundations to support caregivers. It is also more difficult for them to access qualified professionals for the medical care and rehabilitation of people with AD.

Interestingly, respondents who did not declare new needs in relation to Covid-19 indicated that they expected help in the area of providing additional care or support, which we interpreted as needs that have existed for a long time but are still unmet, which is characteristic of the Polish population, where familiarity with care for charges with AD prevails. More than 83% of the caregivers in our study had not been offered help from either family or institutions since the Covid-19 pandemic began. Family caregivers need support in the demanding task of caring for a charge with AD. A study by Kowalska et al. [10] among Polish caregivers a few years earlier showed that 93% of them could rely on help from their family. Interestingly, the results of the 2018 Social Cohesion Survey confirmed that, in Poland, the strongest element of social relations in which older people (65+) functioned was family ties [15]. Data included networks of social contacts, potential sources of help or levels of trust, with family relationships being stronger in rural areas than in cities. A study by Daley et al. [26] highlighted both the value of family support in improving the quality of life by reducing care delivery demands and weight of responsibility, as well as the converse negative impact of lack of family support and conflict on the quality of life.

Our study showed that 75 of the 85 caregivers experienced high level of stress (sten 7–10) in relation to caring for a ward with dementia in the Covid-19 period. The age of the caregivers had no effect on the level of stress experienced. These are worrying findings because chronic stress experienced in relation to caregiving negatively affects mental and somatic health. caregivers are at risk of developing depression, anxiety disorders, sleep disorders, and cognitive decline [15]. Increased burden and negative health consequences for caregivers negatively affect the quality of care and increase the likelihood of early institutionalization of the charge [15,29]. Additionally, the study by Szczepańska-Gieracha et al. [24] showed a significant relationship between the level of depression and the amount of support the caregiver receives.

In an observational study, we examined whether the degree of dementia affected the perceived stress score. Using one-way ANOVA with Scheffé’s post hoc test, we observed a significant difference between the two groups. Interestingly, this occurred between moderate (the lowest scores) and severe degrees (the highest scores). The group declaring that their subjects suffered from a mild form of AD achieved slightly lower scores than the group with a severe form. This is surprising, given the increasing difficulties associated with the nature of the disease. The state of research prior to Covid-19 tended to indicate medium levels of stress in caregivers of charges with dementia. For example, Yu et al. [13] observed average levels of stress in caregivers of charges with mild AD. Additionally, data from Poland indicated average levels of stress in caregivers [10]. Unfortunately, the use of other measurement tools does not allow us to compare our studies.

The progression of dementia totally engages the caregiver. This makes it necessary to devote all of one’s time to care, increases the daily physical effort of the caregiver, and increases the likelihood of sudden stressful events that are difficult to foresee, such as accidents, falls, and fractures. The severity of the disease is one of the most important causes of lower quality of life for the caregiver related to dementia in Alzheimer’s disease and treatment [15]. In view of these results, we decided to take a better look at the group of caregivers who declared a mild degree of dementia of their charge. This group consisted of 20 people (90% women) with an average age of 49 years ± 13.5, mostly married daughters. The duration of the disease was on average 3 years ± 1.9, while the duration of care was also 3 ± 2.4. The mean score on the PSS-10 scale for this group was 25.5 points ± 4.5, while the mean value in stens was 8.

Analyzing the responses in this group, it was clear that in the subjective assessment of the caregiver, the condition of the charge had deteriorated (60% definitely yes; 25% rather deteriorated), and there had been a change in functioning due to Covid-19 (40% definitely yes; 35% rather yes). When it came to assessing the deterioration of one’s own health since the Covid-19 outbreak, the most common response was ‘hard to say’ (35%). Health concerns are an important issue, as 90% of caregivers declared that their concerns about their charge’s health had increased in relation to Covid-19 (60% definitely increased; 30% rather increased). Furthermore, concerns about their own health also increased (40% definitely increased; 30% rather increased). In this group, 80% of the respondents did not receive any help, and 20% received psychological or social help. Our findings of an increase in caregiver concern about and difficulty in assessing their own health in relation to Covid-19 were significant in that caregiver lacked a perception of their own health or neglected it, which is a significant factor in reduced quality of life [26].

These factors contributed to the fact that caregivers in this group have significantly higher levels of stress than caregivers of charges with moderate AD. Exploring the characteristics of a group of caregivers who care for charges with mild AD allowed us to cautiously interpret that high levels of stress may be associated with the onset of multiple problems related to dementia symptoms and caregiving needs. Undoubtedly, caregivers may feel lost in a new situation and stressed by the lack of specialist help. A lack of sound knowledge about the nature of the disease and uncertainty about their ability to care for a charge with AD can affect their perception of their own situation, their level of burnout, and their health. Daley et al. [26] identified a change in the person with dementia as a factor affecting quality of life, with personality changes being the most commonly upsetting factor. The authors reported that some features of the illness, such as repetition and short-term memory loss, were often experienced as frustrating, and anger, poor motivation, and hallucinations as particularly difficult.

Our study revealed the plight of caregivers in relation to the Covid-19 pandemic but also the opportunities for change. We found that the Internet makes it possible to reach out to caregivers who may normally have been excluded and provide them with access to reliable knowledge and psychological support. More than 35% of caregivers said they needed the provision of psychological support during the Covid-19 pandemic, and 19% of caregivers needed the introduction of care-related education programs for AD charges. It seemed that these needs were already noticeable, and the availability of free online advice from psychologists and psychotherapists was increasing. Notably, this is a good area of activity for eHealth (or e-health) interventions, which are defined as healthcare practices delivered via the Internet [30,31]. E-health interventions include psychoeducation, coping strategies/self-management, and social support, as well as remote monitoring, counselling (including decision support), psychosocial therapies, and clinical care [32]. Popularizing this type of intervention could be an excellent response to the needs of caregivers of AD charges during the Covid-19 pandemic.

Observing closed groups for caregivers, one can see how much it means to them to share their problems and how they actively seek social and psychological support. They talk about their loved ones and share their knowledge and experiences, as well as inform others about the death of a charge and thank them for their support. These observations show how important it is to recognize caregivers, with their problems and amazing power. Systemic, free, and professional support is needed to prevent caregiver burnout and to improve the quality of life of caregivers, thereby improving the quality of life of charges with AD.

## 6. Conclusions

The study group was characterized by high levels of stress (mostly 9 sten). The caregiver group of charges with severe AD had the highest level of stress. In most cases, (91.1%), the caregivers of charges with dementia were women. More than 83% of the caregivers reported that they had no offers of help in caring for a ward with AD during the Covid-19. The following were identified as the greatest needs: providing care for the client if the caregiver becomes ill with Covid-19, providing care during working hours, and providing professional psychological support for caregivers.

## Figures and Tables

**Figure 1 ijerph-18-04493-f001:**
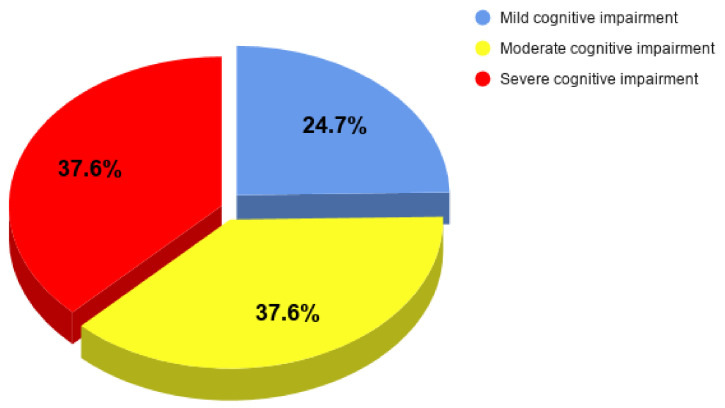
Dementia severity in Alzheimer’s charges before the Covid-19 as assessed by caregivers.

**Figure 2 ijerph-18-04493-f002:**
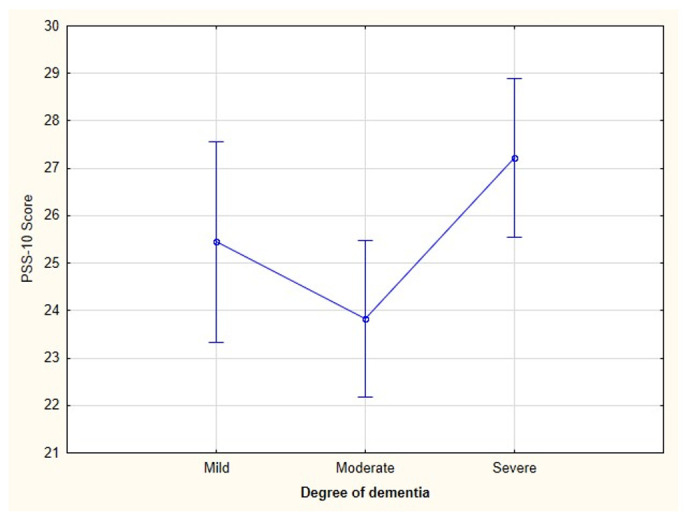
Distribution of sten values on the 10-item Perceived Stress Scale according to the three degrees of dementia in the charges of the caregivers studied. The current effect was F (2.82) = 4.1388 while the *p*-value = 0.01939.

**Table 1 ijerph-18-04493-t001:** Characteristics of caregivers of charges with Alzheimer’s Disease. Abbreviations: SD—standard deviation.

Baseline Characteristics	*n* (%)
Caregivers’ age	
Mean (SD)	51 (± 11.9)
Range	23–78
Caregivers’ gender	
Female	80 (94.1)
Male	5 (5.9)
Caregivers’ relationship to the charge	
Daughter/Son	56 (65.9)
Daughter/Son-in-law	12 (14.1)
Spouse	10 (11.8)
Other	4 (4.7)
Grandchildren	3 (3.5)
Caregiving duration, years	
Mean (SD)	6 (3.7)
Patients’ age	
Mean (SD)	79 (± 8.5)
Range	53–95
Charge illness duration, years	
1–5	44 (51.8)
6–10	33 (38.8)
11–15	6 (7)
16–20	2 (2.4)
Mean (SD)	6.0 (± 3.7)
Residence of the caregivers	
Large city (>300 000)	30 (35.3)
Middle-sized town	11 (12.9)
Small town (≤ 90 000)	22 (25.9)
Village	22 (25.9)
Family life form of the caregivers	
Wife/Husband	60 (70.6)
Miss/Caler	7 (8.2)
Partnership	7 (8.2)
Divorced	7 (8.2)
Separation	2 (2.4)
Widow/Widower	2 (2.4)

**Table 2 ijerph-18-04493-t002:** Covid-19-related problems of caregivers in the daily care of AD charges.

Type	Number of Responses	%
Health care (difficulties in visiting the doctor, buying medicines at the pharmacy)	65	76.5
Difficulties in finding additional care for the person in care	36	42.4
Caring for a family member in care related to the closure of care facilities	29	34.1
Protective measures (use of disinfectant fluids, wearing a mask and gloves)	27	31.8
Difficulties in obtaining psychological assistance/social support	22	25.9
Daily errands (shopping, paying bills, cleaning)	15	17.6
Loss of job or change of job to remote work connected with care of person in care	10	11.8
No new difficulties have emerged	4	4.7

**Table 3 ijerph-18-04493-t003:** Type of assistance expected by caregivers of charges with AD in relation to the Covid-19 pandemic.

Type	Number of Responses	%
Provision of care for the patient with AD if the carer becomes ill with Covid-19	61	71.8
Provision of care for the carer during working hours	46	54.1
Provision of psychological support for the carer during the Covid-19	31	36.5
Providing the possibility to stay in hospital with the patient with AD suffering from Covid-19	27	31.8
Provide legal assistance to the caregiver	25	29.4
Introduction of educational programmes for carers related to functioning during the Covid-19	16	18.8
Provision of financial assistance during the Covid-19	12	14.1

**Table 4 ijerph-18-04493-t004:** Coefficients of correlation of the PSS-10 score with deterioration of illness during Covid-19, caregiver’s health, change in daily functioning, and concerns about the health of the charge with AD in the study group.

Correlation	Deterioration of Illness during Covid-19	Deterioration of Carer’s Health during Covid-19	Change in Daily Functioning	Concerns about the Health of the Charge	Concern for Own Health	PSS-10 Score
Deterioration of illness during Covid-19	1.00	0.39	0.43	0.31	0.16	−0.26
Deterioration of carer’s health during Covid-19	0.39	1.00	0.33	0.21	0.17	-0.53
Change in daily functioning	0.43	0.33	1.00	0.45	0.27	−0.39
Concerns about the health of the charge	0.31	0.21	0.45	1.00	0.60	−0.29
Concern for own health	0.16	0.17	0.27	0.60	1.00	−0.14
PSS-10 score	−0.26	−0.53	−0.39	−0.29	−0.14	1.00

PSS-10—10-item Perceived Stress Scale. Red text indicates that the correlation coefficient is significant at the level of *p* < 0.05.

## Data Availability

The data presented in this study are available on request from the corresponding author. The data are not publicly available due to privacy restrictions.

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
