# Peer review of "Needs of Alzheimer’s Charges’ Caregivers in Poland in the Covid-19 Pandemic—An Observational Study"

_ijerph, 2021, doi:10.3390/ijerph18094493_

Round 1
Reviewer 1 Report
Thank you for the opportunity to review this paper.
This paper is a highly original work and has a significant impact on the field of Alzheimer's disease. This study aimed to identify the needs of the caregivers of people with Alzheimer’s Disease to get extra care and professional psychological support because of the Covid-19 pandemic. The paper is well organized and can be accepted after handling the following points:
- It would be better to modify Table 1. I understand from it that the number of spouses is 56, not 10. Also, the number of daughters and sons is 12. However, you mentioned in the discussion section that most caregivers were children and the second largest group of caregivers was daughter-in-law/son-in-law. So you have to modify it.
- The abbreviation must be written beside the words when it appears for the first time. So, the authors need to check all of them. For example, "AD" is mentioned two times. The first is in the abstract section, while the second time is in the introduction section - page one.
Author Response
SUMMARY OF OUR RESPONSES
Thank you for your careful review of our paper as well as your comments, corrections and suggestions that ensued. A careful revision of the paper has been carried out to take all of them into account, and in the process, we believe the paper has been significantly improved. In the present „Response Letter“ we first detail the major changes that have been made in the paper to correct the main weaknesses identified by the Reviewer. We then sequentially address all the points that we have corrected „step-by-step“.
The main changes:
- Responses to all minor doubts presented by the Reviewers have been given.
- The readability of the table has been improved and the data presented from the highest to the lowest values (%).
- The text was rechecked for linguistic and grammatical correctness.
REVIEWER 1 EVALUATION
Thank you for all your important comments. Our answers to your points are as follows:
It would be better to modify Table 1. I understand from it that the number of spouses is 56, not 10. Also, the number of daughters and sons is 12. However, you mentioned in the discussion section that most caregivers were children and the second largest group of caregivers was daughter-in-law/son-in-law. So you have to modify it.
Response: We agree with this remark. We have modified Table 1 to make it more clear (removing a misleading offset).
The abbreviation must be written beside the words when it appears for the first time. So, the authors need to check all of them. For example, "AD" is mentioned two times. The first is in the abstract section, while the second time is in the introduction section - page one.
Response: Thank you for your astute observation. We have rechecked the paper to ensure that after first presenting the abbreviation, we use it consistently for the rest of the text.
Reviewer 2 Report
The article is very interesting and needed. The issue brings us closer to the situation of carers of people with Alzheimer's.In recognizing the difficulties of carers in fulfilling their role, it would be important to supplement them with regard to having free time and forms of spending this time. This is of great importance in the proper care of the patient, but also in caring for one's own health. Supplementing these facts in this work or inclusion in subsequent works would be advisable. It would make the results more objective.
Author Response
SUMMARY OF OUR RESPONSES
Thank you for your careful review of our paper as well as your comments, corrections and suggestions that ensued. A careful revision of the paper has been carried out to take all of them into account, and in the process, we believe the paper has been significantly improved. In the present „Response Letter“ we first detail the major changes that have been made in the paper to correct the main weaknesses identified by the Reviewer. We then sequentially address all the points that we have corrected „step-by-step“.
The main changes:
- Responses to all minor doubts presented by the Reviewers have been given.
- The readability of the table has been improved and the data presented from the highest to the lowest values (%).
- The text was rechecked for linguistic and grammatical correctness.
REVIEWER 2 EVALUATION
Thank you for all your important comments. Our answers to your points are as follows:
In recognizing the difficulties of carers in fulfilling their role, it would be important to supplement them with regard to having free time and forms of spending this time. This is of great importance in the proper care of the patient, but also in caring for one's own health. Supplementing these facts in this work or inclusion in subsequent works would be advisable. It would make the results more objective.
Response: Thank you very much for appreciating the problem raised and the value of our work. We agree with the reviewer's suggestion to complement the thread on the situation of caregivers in terms of disposition of free time and forms of its spending. Unfortunately, in this work we do not have such a possibility. However, we plan to continue our research and will certainly take this valuable comment into consideration to make the results of the next study more objective. We are convinced that including this aspect of the caregivers’ life will allow us to create a full picture and better understand the difficult situation of caregivers of charges with AD.
Reviewer 3 Report
Definitely, understanding how Covid-19 pandemic has affected caregiving of older adults is extremely important. In this respect, the results given by the authors in their manuscript could be relevant if there were not so predictable. Stress and burnout in caregivers is a well-known and well-documented fact. People with dementia also frequently develop depression, which in many cases can even outgrow dementia onset. Jointly, theoretical background makes finding higher rates of stress and health problems quite predictable and expectable. Of course, authors' results are welcome, but they do not make an important contribution to the state-of-the-art.
The biggest criticism for this paper goes for its methodological conception, realization and analysis. Although experimental conditions clearly explain the necessity to carry out an online test, the test itself has several shortcomings:
- Data collection: collecting data through social networks necessarily conveys the risk to bias the target group. How many older caregivers could have accessed this questionnaire? How many younger caregivers could have accessed it not having accounts in Facebook or other social networks? Certainly, data from social networks are valid for social research, but results must be commented respectively. For instance, how reliable the collected sample is for talking about general caregiver population in Poland? Do not we underestimate that the biggest group of caregivers (= spouses) is not enough represented here?
- Similar comment goes for regional representation. Authors claim to analyze data proceeding from different parts of Poland, including big and small towns and villages. When divided according to this parameter, the sample of 85 participants may result very small. How reliable are the data for inferring about the needs of a specific group of caregivers in this respect?
- Concerning results, I would like to make several observations. First of all, to which extent classification of AD patients into mild, moderate and severe, as made by their caregivers, is reliable? It is quite easy to mix mild and moderate AD, and moderate to severe stages. Furthermore, data analysis would definitely benefit from being presents in grids, since qualitative analysis (as given) really does not allow following data presentation easily. Finally, it seems that not all results are well commented in this section. Maybe, the problem comes from the questionnaire design itself: giving closed answer options may exclude important information.
In Discussion, I really appreciate comments on the differences found between caregivers for Mild AD and Moderate AD.
It seems important that the authors revise the language of the manuscript, since several mistakes and lapsus calami can be observed. There are also several sentences that do not start with a capital letter.
Author Response
SUMMARY OF OUR RESPONSES
Thank you for your careful review of our paper as well as your comments, corrections and suggestions that ensued. A careful revision of the paper has been carried out to take all of them into account, and in the process, we believe the paper has been significantly improved. In the present „Response Letter“ we first detail the major changes that have been made in the paper to correct the main weaknesses identified by the Reviewer. We then sequentially address all the points that we have corrected „step-by-step“.
The main changes:
- Responses to all minor doubts presented by the Reviewers have been given.
- The readability of the table has been improved and the data presented from the highest to the lowest values (%).
- The text was rechecked for linguistic and grammatical correctness.
REVIEWER 3 EVALUATION
Thank you for all your important comments. Our answers to your points are as follows:
- Definitely, understanding how Covid-19 pandemic has affected caregiving of older adults is extremely important. In this respect, the results given by the authors in their manuscript could be relevant if there were not so predictable. Stress and burnout in caregivers is a well-known and well-documented fact. People with dementia also frequently develop depression, which in many cases can even outgrow dementia onset. Jointly, theoretical background makes finding higher rates of stress and health problems quite predictable and expectable. Of course, authors' results are welcome, but they do not make an important contribution to the state-of-the-art.
Response: We disagree with this comment. Indeed, caregiver burnout and stress levels are a well-known fact and present in the literature. However, there is a new factor of living during Covid-19 that has not been considered until now. If we assume that the situation of caregivers is difficult, but do not conduct different studies, the conclusions will remain only speculations. The reviewer found that our work was welcome but did not add anything new to the state of knowledge. Please note the later responses regarding the study group (we were able to reach a younger generation of caregivers, which creates a new perspective), the targeting caregivers from rural areas and small towns, where initiatives for caregivers, support groups, etc. are not available. Equally important is the relationship of caregiver stress levels to the stage of AD in the charge, which are surprising and encourage further research. We believe that the findings of our study regarding the needs of caregivers - psychological and informational support - that are possible through the medium of the Internet are also important. We are aware of the limitations of our study, but this does not mean that our findings are irrelevant.
- The biggest criticism for this paper goes for its methodological conception, realization and analysis. Although experimental conditions clearly explain the necessity to carry out an online test, the test itself has several shortcomings:
Data collection: collecting data through social networks necessarily conveys the risk to bias the target group. How many older caregivers could have accessed this questionnaire? How many younger caregivers could have accessed it not having accounts in Facebook or other social networks? Certainly, data from social networks are valid for social research, but results must be commented respectively. For instance, how reliable the collected sample is for talking about general caregiver population in Poland? Do not we underestimate that the biggest group of caregivers (= spouses) is not enough represented here?
Response: We are aware of the difficulties involved in conducting a survey via the Internet. Please note that we did not only use social networking sites, but also contacted AD carers' foundations, NGOs and AD centers to ensure that the invitation to participate in our study reached as many carers as possible. However, the largest response was obtained from caregiver groups communicating through Facebook, which is also an interesting aspect of this study. We believe that this is related to caregivers seeking social support, help and information about the disease "first-hand", which is apparently achievable through social networks. The study design was somehow forced by the epidemiological situation.
In our article in lines 235-240 we state that the group we studied is not a representative group of caregivers from Poland, but on the other hand its strength lies in the fact that we also managed to reach caregivers from very small settlements as much as those from big cities. The strength of our study is to show a younger generation of caregivers (children, daughter-in-law, son-in-law) that has not been so visible in the available literature before. Let us present this excerpt: “Our study group does not reflect the population of caregivers in Poland, but it does provide a new perspective on the problems of caregivers. We had an almost equal representations of caregivers from large, medium, and small cities, as well as rural areas. This is important because caregivers in smaller towns and rural areas cannot rely on community-based initiatives, NGO-funded activities, and the work of various foundations to support caregivers. It is also more difficult for them to access qualified professionals for the medical care and rehabilitation of people with AD.”
- Concerning results, I would like to make several observations. First of all, to which extent classification of AD patients into mild, moderate and severe, as made by their caregivers, is reliable? It is quite easy to mix mild and moderate AD, and moderate to severe stages. Furthermore, data analysis would definitely benefit from being presents in grids, since qualitative analysis (as given) really does not allow following data presentation easily. Finally, it seems that not all results are well commented in this section. Maybe, the problem comes from the questionnaire design itself: giving closed answer options may exclude important information.
Response: We must also still consider how difficult it is to get a diagnosis and professional help when there is no access to specialists in geriatrics, psychiatry or neuropsychology. In our study, we cared about listening to caregivers and identifying their needs because their situation is unimaginably difficult. The emergence of Covid-19 has exacerbated this situation, which was difficult to imagine even before the epidemic due to the inefficiencies in Polish health care and the insufficient number of programs to support caregivers of people with AD. How reliable is it for caregivers to classify patients into the three stages of AD? We assumed that caregivers know their clients best. In addition, they actively seek out reliable information and experts who can help with diagnosis and treatment, often making them experts on the disease. We asked caregivers to answer honestly, and we assume that the caregivers who chose to participate in our survey answered so on this point as well. Checking documentation to confirm the diagnosis of AD along with the stage would be impossible during a time when interpersonal contact must be kept to a minimum.
The data presentation suggestion is very interesting and we will use it in the continuation of our research.
- Similar comment goes for regional representation. Authors claim to analyze data proceeding from different parts of Poland, including big and small towns and villages. When divided according to this parameter, the sample of 85 participants may result very small. How reliable are the data for inferring about the needs of a specific group of caregivers in this respect?
Response: Thank you for this comment. We understand that this issue may raise some concerns. Our team has been dealing with the problems of caregivers and AD patients in Poland for years. We are professionals who work with caregivers on a daily basis, which allows us to provide a broader perspective. We believe that the data presented are reliable, as they are in line with the data coming from the Central Statistical Office in Poland as well as from previous studies conducted by our team and other researchers in Poland:
- Szczepańska-Gieracha J, Jaworska-Burzyńska L, Boroń-Krupińska K, Kowalska J. Nonpharmacological Forms of Therapy to Reduce the Burden on Caregivers of Patients with Dementia—A Pilot Intervention Study. International Journal of Environmental Research and Public Health. 2020; 17(24):9153. https://doi.org/10.3390/ijerph17249153,
- Kowalska J, Gorączko A, Jaworska L, Szczepańska-Gieracha J. An Assessment of the Burden on Polish Caregivers of Patients With Dementia: A Preliminary Study. American Journal of Alzheimer’s Disease & Other Dementias. December 2017:509-515. doi:1177/1533317517734350,
- Szczepańska-Gieracha J., Kowalska J., Salamon-Krakowska K., Ochnik M., Jaworska-Burzyńska L. Cognitive impairment, depressive symptoms and the efficacy of physiotherapy in elderly people undergoing rehabilitation in a nursing home facility. Advances in Clinical and Experimental Medicine, 2010, 2010 : vol.19, nr 6, s.755-764.
- Mazurek J, Kowalska J, Rymaszewska J. Psychogeriatric care in Poland. Geriatric Mental Health Care. 2013; 1(1): 7-10.
- Rachel W, Jabłoński M, Datka W, Zięba A. Caregivers’ health related quality of life in Alzheimer’s disease. Psychogeriatria Polska. 2014; 11(3):67-78.
- In Discussion, I really appreciate comments on the differences found between caregivers for Mild AD and Moderate AD.
Response: Thank you for this comment.
- It seems important that the authors revise the language of the manuscript, since several mistakes and lapsus calami can be observed. There are also several sentences that do not start with a capital letter.
Response: Taking care of clarity and scientific value of our work we also took the effort of professional, certified proofreading. Naturally, after receiving the reviewer's comment, we carefully checked the text of the paper and corrected the questionable parts.
Reviewer 4 Report
The topic of this manuscript is quite important, as the burden for caregivers of people with dementia is very high, and more with the impact of COVID-19 pandemic.
The work is quite simple but it corroborates some problems that we suspected that were present in the care of people with dementia
Some things to comment.
- The manuscript refers all the time to Alzheimer’s disease, but from what is indicated, it is not clear that the patients of this study had a diagnosis of this dementia specifically. If there was not a conclusive diagnose, I think that instead of Alzheimer, the authors should use the term dementia.
- How the participants of the study were contacted or chosen?
- Figure 1. It could be more illustrative if it included the data before and after COVID-19
- I suggest that in tables 1 and 2, to present the types ordered from higher to lower %.
Author Response
SUMMARY OF OUR RESPONSES
Thank you for your careful review of our paper as well as your comments, corrections and suggestions that ensued. A careful revision of the paper has been carried out to take all of them into account, and in the process, we believe the paper has been significantly improved. In the present „Response Letter“ we first detail the major changes that have been made in the paper to correct the main weaknesses identified by the Reviewer. We then sequentially address all the points that we have corrected „step-by-step“.
The main changes:
- Responses to all minor doubts presented by the Reviewers have been given.
- The readability of the table has been improved and the data presented from the highest to the lowest values (%).
- The text was rechecked for linguistic and grammatical correctness.
REVIEWER 4 EVALUATION
Thank you for all your important comments. Our answers to your points are as follows:
The manuscript refers all the time to Alzheimer’s disease, but from what is indicated, it is not clear that the patients of this study had a diagnosis of this dementia specifically. If there was not a conclusive diagnose, I think that instead of Alzheimer, the authors should use the term dementia.
Response: Thank you for your insights. When engaging caregivers for our study, we invited caregivers whose charges suffer from Alzheimer's Disease. Due to the specific design of our study, we were not able to verify that the charges were actually diagnosed with AD, so we trusted the caregivers. In this case, we address an important issue, which is the difficulty of obtaining a diagnosis in Poland due to the inefficiency of the health care system (we described this in the introduction section). We also took into account the fact that AD accounts for 60-70% of all dementias. Therefore, we relied on the knowledge and competence of the caregivers who decided to participate in our study. In summary, we decided that a consequence flowing from the design of our study was to use the term Alzheimer's Disease instead of the broader term dementia.
How the participants of the study were contacted or chosen?
Response: We began the process by sending out information about the study via email and social networking websites for caregivers of people with AD across the country, including open and private groups for caregivers on Facebook, portals and news sites dedicated to dementia and elder care. An invitation to participate in the study was also sent via email to organizations of caregivers of residents with AD with a request to disseminate information to them. Group selection was inherently random. We included inclusion criteria in the invitation, which were: providing care for AD charges before and during the Covid-19 pandemic; completion of an online questionnaire assessing the situation of caregivers and the needs of care during a pandemic, and consent to participate in the study, which means assessing the stress level using the 10-item Perceived Stress Scale.
Figure 1. It could be more illustrative if it included the data before and after COVID-19
Response: We agree with the comment. Unfortunately, we only have information about what stage of disease the charge was in prior to Covid-19 and whether his condition has deteriorated or not. Deterioration of the AD charge's health during Covid-19 was declared by 53% of caregivers. Of course, this does not mean that the stage of the disease has changed.
I suggest that in tables 1 and 2, to present the types ordered from higher to lower %.
Response: Thank you very much for your excellent comment, which will improve the readability of the presented data.